# Impact of Cryopreservation of Peripheral Blood Stem Cells (PBSC) in Transplantation from Matched Unrelated Donor (MUD)

**DOI:** 10.3390/jcm11144114

**Published:** 2022-07-15

**Authors:** Gabriele Facchin, Chiara Savignano, Marta Lisa Battista, Miriam Isola, Maria De Martino, Giuseppe Petruzzellis, Chiara Rosignoli, Umberto Pizzano, Michela Cerno, Giulia De Cecco, Antonella Bertone, Giovanni Barillari, Renato Fanin, Francesca Patriarca

**Affiliations:** 1Division of Hematology and Stem Cell Transplantation, Azienda Sanitaria Universitaria Friuli Centrale, Piazzale S. Maria della Misericordia 10, 33100 Udine, Italy; martalisa.battista@asufc.sanita.fvg.it (M.L.B.); giuseppe.petruzzellis93@gmail.com (G.P.); chiara.rosignoli@edu.unife.it (C.R.); pizzano.umberto@spes.uniud.it (U.P.); michela.cerno@asufc.sanita.fvg.it (M.C.); renato.fanin@asufc.sanita.fvg.it (R.F.); 2Department of Medical Area (DAME), University of Udine, Via Colugna 50, 33100 Udine, Italy; miriam.isola@uniud.it (M.I.); maria.demartino@uniud.it (M.D.M.); 3Department of Transfusion Medicine, Azienda Sanitaria Universitaria Friuli Centrale, Piazzale S. Maria della Misericordia 10, 33100 Udine, Italy; chiara.savignano@asufc.sanita.fvg.it (C.S.); giulia.dececco@asufc.sanita.fvg.it (G.D.C.); antonella.bertone@asufc.sanita.fvg.it (A.B.); giovanni.barillari@asufc.sanita.fvg.it (G.B.); 4Division of Medical Statistic, Department of Medicine, University of Udine, P.le Kolbe n. 4, 33100 Udine, Italy

**Keywords:** cryopreservation, allogeneic stem cell transplantation, matched unrelated donor, GvHD, engraftment, peripheral blood stem cells

## Abstract

**Background**: Cryopreservation of PBSC for allogenic hematopoietic stem cell transplantation (allo-HSCT) was implemented due to the current Coronavirus 2019 pandemic. The impact of match unrelated donor (MUD) graft freezing on the outcome of allo-HSCT in terms of hematological recovery, graft versus host disease (GVHD), and survival are still controversial. **Methods**: In this study, we compared graft composition, clinical characteristics, and outcome of 31 allo-HSCT from MUD cryopreserved PBSC (Cryo Group) with 23 matched-pair allo-HSCT from fresh MUD PBSC (Fresh Group) performed in our center between January 2020 and July 2021. **Results**: No significant differences were recognized in clinical characteristics of patients, donors, and transplants between the Cryo and Fresh groups except for a better prognostic comorbidity index (HCT-CI) of the Cryo group. In the Cryo Group, the median time from apheresis to cryopreservation was 46.0 h (range 23.8–53.5), while the median time from cells collection and reinfusion was 13.9 days (range 5.8–28.1). In the Fresh Group, median time from apheresis to reinfusion was 35.6 h (range 21.4–51.2). The number of viable (7-AAD negative) CD34+ cells per kg patient infused was significantly lower in the Cryo Group (5.2 ± 1.9 × 10^6^/kg vs. 7.0 ± 1.3 × 10^6^/kg; *p* < 0.001). Indeed, there was a 36% (11–70) median loss of viable CD34+/kg cells after freezing. All patients engrafted: median time to neutrophil engraftment (>0.5 × 10^9^/L) was 13.5 days (range 12–15) for Cryo Group and 14 days (range 13–16) days for Fresh Group (*p* = 0.522), while the median time to platelet engraftment (>20 × 10^9^/L) was, respectively, 14 (range 12–18) and 15 (range 12–17) days (*p* = 0.904). The incidence of grade ≥ 2 acute GVHD was similar in the two groups (56.5% Cryo Group vs. 60.0% Fresh Group; *p* = 0.832) and no differences in terms of OS (*p* = 0.090), PFS (*p* = 0.200) and TRM (*p* = 0.970) were observed between the Cryo and Fresh groups. **Conclusions:** In our series, no differences between the Cryo and Fresh groups were found in engraftment, grade ≥ 2 acute GVHD incidence, OS, PFS, and TRM despite a lower CD34+ infused dose in the Cryo Group. Frozen PBSCs could be considered a safe option also for allo-HSCT from MUD but a higher amount of PBSC should be collected to warrant an adequate viable CD34+ post-thawing.

## 1. Introduction

Prior to the COVID-19 pandemic, donor grafts in allogeneic hematopoietic stem cell transplantation (allo-HSCT) were usually administered fresh [1,2,3]. The ongoing COVID-19 pandemic caused significant logistical difficulties in the management of allo-HSCT from unrelated donors. Goods and people transport limitations as well as the risk of SARS-CoV-2 community-acquired infection of donors during stem cell mobilization and collection prompted the main international scientific societies and regulatory authorities to recommend the cryopreservation of stem cells to ensure the safety of transplant procedures [4,5]. However, while the experience related to the cryopreservation of HSC is well established in the autologous and cord blood source [6,7,8], in the literature there are a few controversial data regarding allogeneic grafts, especially in the unrelated donors setting [1,2,3,9,10,11,12,13,14]. An important aspect to consider, especially in the MUD setting, is the time from collection to cryopreservation and infusion. The effects of transit, holding, and processing times on transplant safety and efficacy are not well known.

To better understand the impact of cryopreservation on post-transplant outcomes in a PBSC MUD HSCT setting, we compared a cohort of MUD HSCT from cryopreserved PBSC (Cryo Group) with another cohort from fresh grafts (Fresh Group) performed at our center during the COVID-19 pandemic during the same period.

## 2. Methods

### 2.1. Study Cohort

We consider all adult patients (age ≥ 18 years old) who underwent first allo-HSCT with MUD PBSC grafts between January 2020 and July 2021. A total of 56 patients were included, 32 of them in the Cryo Group and 23 in the Fresh Group. Cryopreservation of grafts was made according to recommendations from international scientific societies and regulatory authorities during pandemic phases [4,5] especially in cases of no backup donor and/or long distance from donor center, while outside pandemic waves or when backup donors were available fresh products were requested. One patient from the Cryo Group was not infused because of a pre-transplant diagnosis of metastatic melanoma. This patient was not considered for statistical analysis (Cryo Group n = 31; Fresh Group n = 23).

### 2.2. Cryopreservation Procedure

All grafts were collected from peripheral blood after mobilization according to standard procedures based on subcutaneous injection of GCSF routinely administered for 4 to 5 days. Products were transported to our center under controlled rate temperature conditions (4 ± 2 °C). Once the graft arrived at the tissue establishment, it was subjected to a quality control (QC) test [AB0 testing, sterility, total nucleated cells (TNC) CD34+ and CD3+ cells count] and minimal manipulation consisting of centrifugation and/or dilution with 5% albumin solution to obtain a final concentration of about 100 × 10^6^ TNC/mL. Product processing was made after overnight storage depending on time of arrival. The product was infused with no additional manipulation in the Fresh Group. In the frozen group, cryopreservation was performed according to standard operating procedures at our center. Cryoprotectant solution containing 5% Serum Albumin and 20% dimethylsulfoxide (DMSO) was slowly added to cellular suspension at 0–4 °C temperature (final DMSO concentration 10%). The cell suspension was then divided into 6–8 cryo-bags, cryopreserved by controlled rate freezing device, and finally stored in vapor phase liquid nitrogen cryostorage tank (<−160 °C).

At the time of reinfusion, the cryopreserved products were thawed rapidly in a warm water bath at 37 °C and infused. TNC and CD34+ cell viability in Cryo Group was evaluated with Trypan blue (TB) and 7-Aminoactinomycin D (7-AAD) before cryopreservation, on satellite tube (after 48 h from freezing procedure), and at the time of reinfusion. Same viability tests were performed at the arrival and at the reinfusion in the Fresh Group.

### 2.3. Definitions

Neutrophil recovery was defined as the date of the first of three consecutive days with an absolute neutrophil count > 0.5 × 10^9^/μL and platelet recovery was defined as the date of the first platelet count > 20 × 10^9^/μL without transfusion for the last 7 days.

Diagnosis and grade of acute graft versus host disease (aGVHD) and chronic GVHD (cGVHD) were documented using international standard criteria [15].

The hematopoietic cell transplantation-specific comorbidity index (HCT-CI) was calculated for all patients.

Progression-free survival (PFS) was defined as the time from stem cell infusion to disease relapse, progression, or death from any cause. Transplant-related mortality (TRM) was defined as death due to all causes not related to the underlying hematological disease, while overall survival (OS) was defined as the time from transplantation to either death or last follow-up of the patients.

Time from apheresis to cryopreservation was considered from the start of the apheresis process to the end of cryopreservation and includes time for apheresis procedure, transit time, overnight storage (if present), and time for cryopreservation process.

### 2.4. Study Endpoints and Statistical Analysis

The primary study endpoint was time for neutrophil and platelet engraftment. Secondary endpoints were incidences of grades ≥ 2 aGHVD, OS, PFS, and TRM. Data were collected in an XLS Database (Microsoft office 2016; Microsoft Corporation, Redmond, WA, USA). Absolute values, percentages, mean and median (standard deviation (SD) or interquartile range (IQR)) were calculated. Categorical variables were compared using the chi-squared test or Fisher’s exact test, while continuous variables were compared using a Student *t*-test or Mann–Whitney U test, according to the Shapiro–Wilk test establishing whether data were normally or non-normally distributed. Univariable and multivariable linear regression was performed to estimate the association between the time to neutrophil and platelet recovery and clinical/demographic variables, by calculating the β coefficient and 95% confidence intervals (CIs). The multivariable analyses included all variables significant at *p* < 0.10 in the univariable analysis, considering potential collinearities. Time to neutrophil and platelet engraftment, overall survival (OS), and progression-free survival (PFS) were calculated according to Kaplan–Meier method, and log-rank tests were used to compare Fresh and Cryopreserved groups. The cumulative incidence method was used to estimate TRM accounting for the presence of competing risks. A *p*-value < 0.05 was considered statistically significant. Analyses were performed using STATA 17.0.

## 3. Results

### 3.1. Baseline Characteristics of Patients, Donors, and Grafts

A total of 54 patients were included, of them, 31 were in the Cryo Group and 23 in the Fresh Group. Patient characteristics are summarized in Table 1. The median age at transplant was 56 (43–63) and 61 (57–65) years, respectively (*p* = 0.180). There were no significant differences between cohorts in terms of gender (*p* = 0.253), disease status at HCT (*p* = 0.361), and performance status (Karnofsky *p* = 0.169; ECOG *p* = 0.056). However, the proportion of patients with HCT-CL > 2 is higher in the Fresh Group (16.7% vs. 47.8%, *p* = 0.047) even if the performance status was well balanced.

The median donor age was 29 (23–36) years in the Cryo Group and 29 (22–33) years in the Fresh Group. AB0 major donor/recipient mismatch, donor/recipient sex mismatch, HLA compatibility, and donor/recipient weight discrepancy were comparable between the two cohorts. The conditioning regimen and GVHD prophylaxis were similar in both the Cryo and Fresh Group.

The median time from apheresis to cryopreservation in the Cryo Group was 46.0 h (range 23.8–53.5) while the median time from cells collection and reinfusion was 13.9 days (range 5.8–28.1). In the Fresh Group, the median time from apheresis to reinfusion was 35.6 h (range 21.4–51.2). The mean number of viable (7-AAD negative) CD34+ cells per kg patient infused were significantly lower in the Cryo Group (5.2 ± 1.9 × 10^6^/kg vs. 7.0 ± 1.3 × 10^6^/kg, *p*< 0.001). In one case, from the Cryo Group, the infused CD34+ cell rate was lower than 2 × 10^6^/kg (minimum amount for safe engraftment according to published data) [16,17,18], because there was a loss of 70% of viable CD34+ cells after cryopreservation.

### 3.2. Clinical Outcome

Median follow-up was 13 months (range 8–14) in the Cryo Group vs. 17 months (range 12–19 in the Fresh Group (*p* = 0.043). Even if there was a 36% median loss of viable CD34+/kg cells after freezing, all patients engrafted. Median time to neutrophil engraftment (>0.5 × 10^9^/L) was 13.5 days (range 12–15) for the Cryo Group and 14 days (range 13–16) for the Fresh Group (*p* = 0.522), while median time to platelet engraftment (>20 × 10^9^/L) was, respectively, 14 (range 12–18) and 15 days (range 12–17) (*p* = 0.904) (Figure 1).

Median time to reach PMN > 1000/mmc was 14 days (range 13–16) for both groups, while the time to reach platelets > 50 × 109/L was 19 (range 15–24) for the Cryo Group and 18 days (15–20) for the Fresh Group (*p* = 0.904; *p* = 0.469). Transfusion blood and platelet support were similar between the Cryo and Fresh groups (*p* = 0.839; *p* = 0.863). No differences in aGVHD incidence, grade, and response to first-line treatment were observed in the two cohorts as shown in Table 2, while chronic GVHD was more frequent in the Fresh Group (34.8% vs. 6.5%, *p* = 0.012). On univariate analysis, time to neutrophil > 0.5 × 10^9^/L, neutrophil > 1.0 × 10^9^/L, platelet > 20 × 10^9^ and platelet > 50 × 10^9^/L were not influenced by the cryopreservation and viable CD34+/kg infused.

The one-year PFS was 71.0% (95% confidence interval [CI] 51.6–83.7) in patients transfused with cryopreserved grafts and 65.2% (95% CI 42.3–80.8) in patients transplanted with fresh grafts (*p* = 0.200) (Figure 2). 

The corresponding rates of OS were 80.7% (95% CI 61.9–90.8) and 78.3% (95% CI 55.4–90.3) in the Cryo and Fresh groups, respectively (*p* = 0.090) (Figure 3).

The one-year cumulative incidence of TRM was 13.0% (95% CI 4.1–27.2) and 13.5% (95% CI 3.4–30.7), respectively (Figure 4).

The only previously mentioned case that received less than 2 × 10^6^/Kg cryopreserved PBSC showed no differences in terms of engraftment and clinical outcome.

## 4. Discussion

This real-life analysis reports our experience during the COVID-19 pandemic with PBSC unrelated donor grafts. Our results suggest that cryopreservation of MUD PBSC during the COVID-19 pandemic, despite a median loss of 36% of viable CD34+ cells, did not negatively affect clinical outcomes.

The Cryo and Fresh groups were well balanced considering baseline characteristics of patients, donors, and grafts. Even if a proportion of patients with HCT-CL > 2 was higher in the Fresh Group (16.7% vs. 47.8%, *p* = 0.047), performance status was similar in the two cohorts.

A DKMS stem cell donor registry analysis concluded that 5–10% of cryopreserved products are not transfused and previous studies have shown also a higher rate of non-transfused graft [5,19]. The main reasons for non-transplantation include relapse/progression of underlying disease, clinical conditions of patients, and dissatisfaction with quality status of the product after the thawing [5,19]. In our experience, only one patient (1/32, 3%) was not infused, due to the diagnosis of metastatic melanoma during screening exams before the transplant procedure.

The effects of transit, holding, and processing times on transplant safety and efficacy are not well known and conflicting data are reported in the literature. Alotaibi A et al. [9] in a large retrospective study including 958 allo-HSCT from 2010 to 2018, reported that graft cryopreservation did not affect engraftment or survival. Hamadani et al. reported similar data analyzing 274 HSCT using cryopreserved grafts with post transplantation cyclophosphamide as GVHD prophylaxis [20]. A limit of these studies was that only a small number of transplant procedures were performed with cryopreserved grafts from unrelated donors where transit, holding, and processing time are more relevant for cellular viability. Indeed, the largest retrospective study from the Center for International Blood and Marrow Transplant Research registry examining the effect of cryopreservation on HCT outcomes, underlined that cryopreservation was associated with inferior outcomes in unrelated PBSC grafts in contrast to related PBSC grafts [3]. Recent studies also reported a risk of graft failure between 3 and 13% in cryopreserved MUD PBSC [3,5,21]. In contrast, Fernandez-Sojo et al. compared 32 patients who underwent matched unrelated donor (MUD)-HSCT using cryopreserved PBSC with 32 patients who underwent MUD-HSCT using fresh grafts during the COVID-19 pandemic: despite a lower dose of CD34+ viable cells infused, no differences were observed regarding engraftment, GVHD, TRM, PFS, and OS between two cohorts. In our series, all patients were engrafted with no cases of primary or secondary graft failure both in the Cryo and Fresh groups, suggesting that freezing procedures do not negatively impact the engraftment capacity of PBSC. Despite a significant loss of viable CD34+ cells (36%) with freezing procedures, there were no differences in terms of time to PMN and PLT engraftment (Figure 1). This data indirectly suggests that there was no effect of cryopreservation on PBSC engraftment potential. This is in contrast with Liznov M. et al. who suggested that PBSC becomes more sensitive to cryopreservation after transport with a consequent higher risk of graft failure [10]. We did not perform functional assays on grafts but in our series only one patient received less than 2 × 10^6^/kg CD34+, while Liznov et al. reported a higher rate of patients that received PBSC under the threshold to ensure safe engraftment, justifying a rate of graft failure higher than ours. These results should advise clinicians to take into account the significant CD34+ cell loss during cryopreservation in the HSCT planning and in the donor selection to ensure an adequate cell dose.

In the literature, there are only a little data regarding the impact of cryopreservation on lymphocyte function. In our series, the CD3+/kg infused dose was not statistically different between the Cryo and Fresh groups (*p* = 0.087) suggesting that lymphocytes are less sensitive to the negative effect of cryopreservation. Unfortunately, data regarding subsets of infused lymphocytes were not available. Some authors suggest that cryopreservation influences the expression of several antigens on lymphocytes and that this phenomenon contributes to the GVHD mechanism [9,22,23]. Previous studies demonstrated similar rates of aGVHD between cryopreserved and fresh grafts [9,12,13], as we confirm in our series in terms of aGVHD incidence, grade, stage, involved organs, or steroid response suggesting that changes in lymphocyte antigen expression do not impact acute GVHD. Chronic GVHD (cGVHD) was more frequent in the Fresh Group (34.8% vs. 6.5%, *p* = 0.012). These data are in contrast with those previously reported by Alotaibi et al. [9]; however, in our study, median follow-up was longer in the Fresh Group (17 vs. 13 months, *p* = 0.043), so probably the incidence of cGVHD may be overestimated compared to the Cryo Group. A longer follow-up may be needed to analyze the impact of cryopreservation on chronic GVHD incidence and characteristics as well as the effects on immune recovery.

The one-year cumulative incidence of TRM was not different between the Cryo and Fresh groups in our series, such as OS and PFS, suggesting that cryopreservation does not impact the safety of the transplant procedure. The previous analysis by Eapen et al. [24] reported higher 1-year mortality in patients receiving cryopreserved bone marrow (BM) grafts for aplastic anemia probably due to the higher sensitivity of the BM product to cryopreservation. Indeed, our results are in line with Jacob et al. including only PBSC, and with Hamadani et al. considering mainly PBSC followed by post-transplantation cyclophosphamide [20,25].

We acknowledge some study limitations such as the retrospective single-center design, the small cohort of patients, the short follow-up, and the lack of information regarding graft composition and immune recovery. However, a point of strength of our single-center study is the standardization of cryopreservation procedures and controls that are still quite heterogeneous in clinical practice, as reported by Lanza et al. in a recent Italian survey [7].

## 5. Conclusions

In our series, no differences between the Cryo and Fresh groups were found in short-term clinical outcomes (engraftment, acute GVHD, OS, PFS, and TRM), despite a lower CD34+ infused dose in the Cryo Group. Further, multi-center prospective studies and a longer follow-up are needed to better understand the clinical impact of cryopreservation on MUD HSCT.

## Figures and Tables

**Figure 1 jcm-11-04114-f001:**
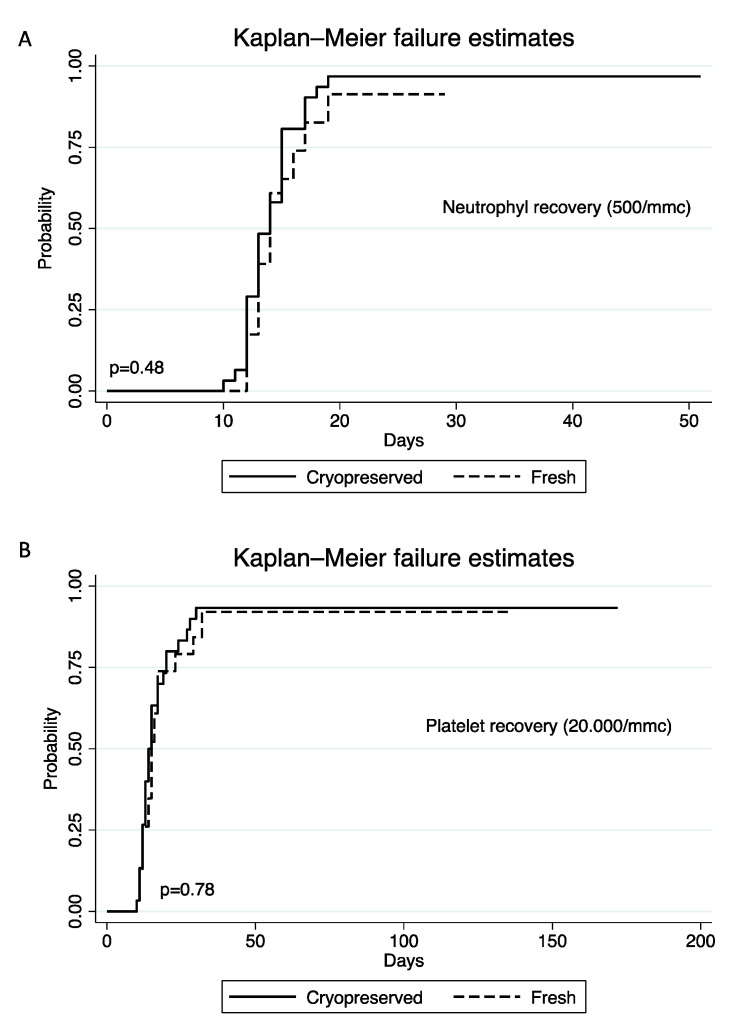
Cumulative incidence of neutrophil (**A**) and platelet (**B**) engraftment in Cryo and Fresh groups.

**Figure 2 jcm-11-04114-f002:**
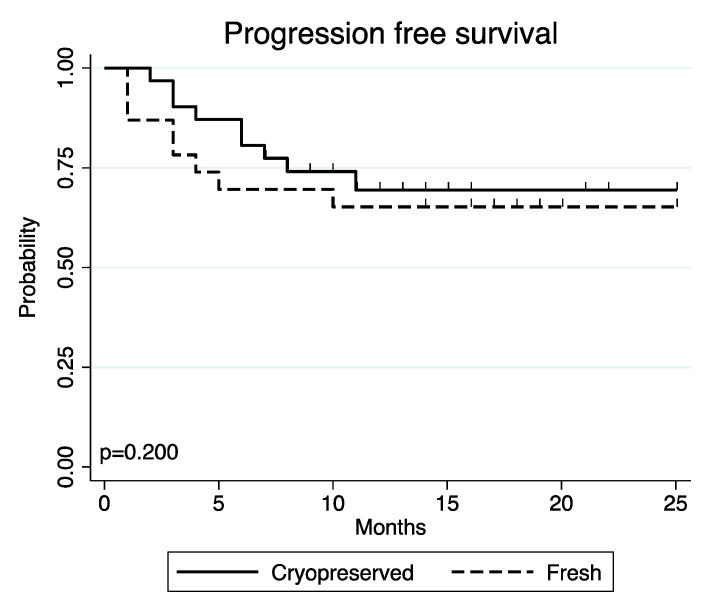
PFS in patients receiving cryopreserved vs. fresh grafts.

**Figure 3 jcm-11-04114-f003:**
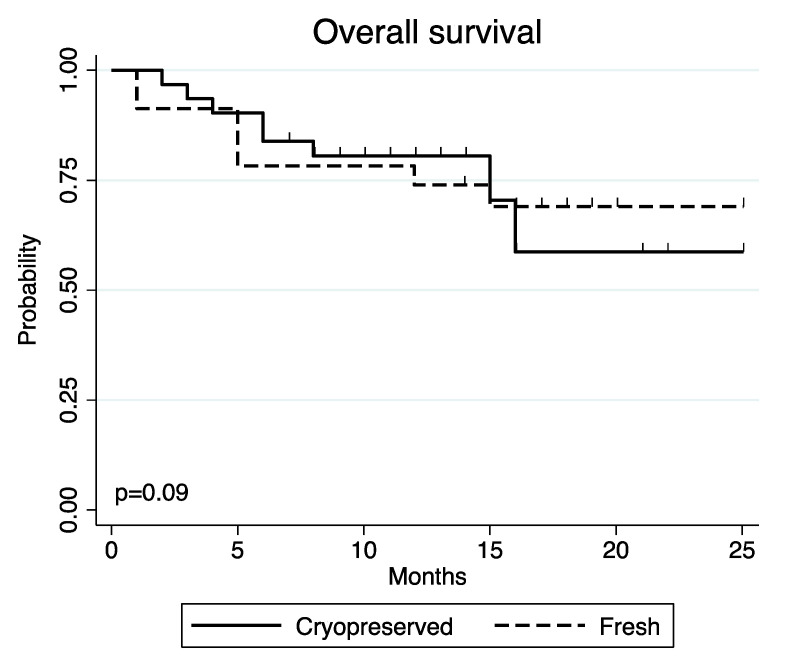
OS in patients receiving cryopreserved vs. fresh grafts.

**Figure 4 jcm-11-04114-f004:**
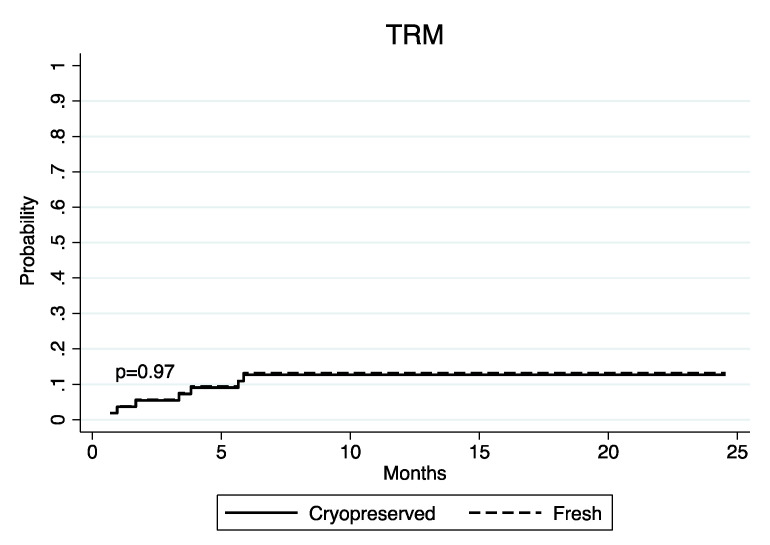
Cumulative incidence of TRM in Cryo and Fresh groups.

**Table 1 jcm-11-04114-t001:** Patients’ characteristics.

	Cryopreserved	Fresh	*p* Value
	(n = 31)	(n = 23)	
**AGE at transplant**, median (IQR)	56 (43–63)	61 (57–65)	0.18
**Sex**, n (%)			0.253
- Male	14 (45.2)	14 (60.9)	
- Female	17 (54.8)	9 (39.1)	
**Patient Weight**, median (IQR)	77 (60–85)	74 (61–89)	0.696
**Disease**, n (%)			0.524
- AML **	13 (41.9)	14 (60.9)	
- ALL	9 (29.0)	3 (13.0)	
- MDS	1 (3.2)	0 (0)	
- NHL	4 (12.9)	3 (13.0)	
- Other	4 (12.9)	3 (13.0)	
**Disease Status**, n (%)			0.361
- CR ***	16 (51.6)	16 (69.6)	
- R/R	10 (32.3)	4 (17.4)	
- RP	3 (9.7)	3 (13.0)	
- SD	2 (6.4)	0 (0)	
**HCT-CL**, n/N (%)			**0.047**
- 0	18/30 (60.0)	8 (34.8)	
- 1–2	7/30 (23.3)	4 (17.4)	
- >2	5/30 (16.7)	11 (47.8)	
**Karnofsky**, n (%)			0.169
- <90%	13 (41.9)	14 (60.9)	
- >90%	18 (58.1)	9 (39.1)	
**ECOG**, n(%)			0.056
- ≥2	27 (87.1)	15 (65.2)	
- <2	4 (12.9)	8 (34.8)	
**AB0**, n (%)			0.15
- Major incompatibility	13 (41.9)	6 (26.1)	
- Minor incompatibility	12 (38.7)	7 (30.4)	
- Compatible	6 (19.4)	10 (43.5)	
**Donor Age**, median (IQR)	29 (23–36)	29 (22–33)	0.964
**Donor/Recipient Sex Mismatch**, n (%)	13 (41.9)	9 (39.1)	0.836
**Donor/Recipient Weight** **Discrepancy (>10 Kg body weight)**, n (%)	19 (61.3)	18 (78.3)	0.184
**HLA**, n (%)			0.399
- 10/10	21 (67.7)	13 (56.5)	
- others	10 (32.3)	10 (43.5)	
**Condition Regimen**, n (%)			0.393
- Myeloablative	21 (67.7)	18 (78.3)	
- Reduce intensity	10 (32.3)	5 (21.7)	
**Patient/Donor CMV Status**, n/N (%)			0.727
- −/−	5/30 (16.7)	1/19 (5.3)	
- −/+	3/30 (10.0)	2/19 (10.5)	
- +/+	10/30 (33.3)	8/19 (42.1)	
- +/−	12/30 (40.0)	8/19 (42.1)	
**WBC/kg Infused**, mean ± SD	9.7 ± 7.1	7.8 ± 2.7	0.217
**CD34/kg Infused**, mean ± SD	5.2 ± 1.9	7.0 ± 1.3	**<0.001**
**CD3/kg Infused**, mean ± SD	15.3 ± 5.1	20.8 ± 10.3	0.087
**Days to neutrophil recovery (500/mmc)**, median (IQR)	13.5 (12–15)	14 (13–16)	0.522
**Days to neutrophil recovery (1000/mmc)**, median (IQR)	14 (13–16)	14 (13–16)	0.904
**Days to platelet recovery (20,000/mmc)**, median (IQR)	14 (12–18)	15 (12–17)	0.744
**Days to platelet recovery (50,000/mmc)**, median (IQR)	19 (15–24)	18 (15–20)	0.469

** AML = acute myeloid leukemia; ALL = acute lymphoid leukemia; NHL = non-Hodgkin Lymphoma; *** CR = complete remission; R/R = relapsed/refractory.

**Table 2 jcm-11-04114-t002:** Acute GVHD characteristics.

	Cryopreserved	Fresh	*p* Value
	(n = 31)	(n = 23)	
**Acute GVHD**, n (%)			
- **Yes**	23 (74.2)	15 (65.2)	
**Grade**, n/N (%)			0.475
- I	10/23 (43.5)	6/15 (40.0)	0.832
- II–IV	13/23 (56.5)	9/15 (60.0)	
**Days onset**, median (IQR)	24 (17–37)	28 (23–45)	0.464
**Skin involvement**, n (%)			
- Yes	21 (67.7)	15 (65.2)	0.846
**Grade**, n/N(%)			0.2
- I	6/21 (28.6)	1/15 (6.7)	
- II–IV	15/21 (71.4)	14/15 (93.3)	
**Gut involvement**, n (%)			
- Yes	7 (22.6)	5 (21.7)	0.941
**Grade**, n/N (%)			0.417
- I	7/7 (100)	4/5 (80.0)	
- II	0/7 (0)	1/5 (20.0)	
**Hepatic involvement**, n (%)			0.502
- Yes	2 (6.4)	0 (0)	
**Grade**, n/N (%)			
- II	2/2 (100)		
**Treatment**, n/N (%)			1
- Steroid	21 (91.3)	14/16 (87.5)	
- observation	2 (8.7)	2/16 (12.5)	
**Response to first line treatment**, n/N (%)			0.43
- Complete response	15 (65.2)	11/14 (78.6)	
- Partial response	2 (8.7)	2/14 (14.3)	
- Refractory	6 (26.1)	1/14 (7.1)	

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
