# Peer review of "Impact of Cryopreservation of Peripheral Blood Stem Cells (PBSC) in Transplantation from Matched Unrelated Donor (MUD)"

_jcm, 2022, doi:10.3390/jcm11144114_

Round 1

Reviewer 1 Report

Dear authors,

Congratulations for your work.  It is also important to review our results, although international reviews have already done so by this time. 

I would highly suggest to have someone to review the language because there are many minor mistakes. It would also be important to edit your tables to be easier to be read.

In the conclusion, it would be appropriate to acknowledge that the study was not powered to detect diferences but a convenience sample to reflect your experience during the Covid pandemic.

Congratulations.

Author Response

Dear reviewer 1,

Thanks for your suggestions. Here the point-by-point response

Point 1: I would highly suggest to have someone to review the language because there are many minor mistakes. It would also be important to edit your tables to be easier to be read.

Response 1: we reviewed language and edit tables. 

Point 2: In the conclusion, it would be appropriate to acknowledge that the study was not powered to detect diferences but a convenience sample to reflect your experience during the Covid pandemic. 

Response 2: we acknowledge that point at the beginning of discussion paragraph (see line 191 of the manuscript: “This real-life analysis reports our experience during COVID-19 pandemic with PBSC unrelated donor grafts.”)

Reviewer 2 Report

Your article about the impact of cryopreservation in PBSC transplantation from Matched Unrelated Donor (MUD) is a well written article providing more evidence to known facts about the cryopreservation. Please find below the suggestions for re-exploration your results and add to your article. 

- Explanations for the time period from apheresis to cryopreservation would make the readers understand the process better. Also, storage, transportation conditions could be stretched more.

- Except the 10% DMSO, no details for the cryoprotectant solution is provided, which is very important for cryopreservation and thawing.

- In methods part, in 2.1. Study cohort section it has been stated that "cryopreservation of grafts made according to recommendations from international scientific societies and regulatory authorities", but there are no references.

- One wonders if to analyze the difference between the myeloablative and reduced intensity group, and also between the different ABO groups could provide ant additional information or not.

- There is a significant p-value for HCT-CL which has not been discussed in discussion part, this should be discussed in relation to other parameters. 

Author Response

Dear reviewer 2,

thanks for your suggestions! Here the point-by-point response:

Point 1: Explanations for the time period from apheresis to cryopreservation would make the readers understand the process better. Also, storage, transportation conditions could be stretched more. 

Response 1: We implemented the section 2.2 Cryopreservation procedure (see line 70-80 of the revised manuscript uploaded):“Products were transported to our Center under controlled rate temperature conditions (4 ± 2°C).”

We also implement paragraph 2.3 Definitions (see line 111-113 of the revised manuscript uploaded): Time from apheresis to cryopreservation was considered from the start of apheresis process to the end of cryopreservation and includes time for apheresis procedure, transit time, overnight storage (if present) and time for cryopreservation process.

Point 2: Except the 10% DMSO, no details for the cryoprotectant solution is provided, which is very important for cryopreservation and thawing.

Response 2: we detailed better the section 2.1 “Cryopreservation procedure” (see line 85-91 of the revised manuscript uploaded): In the frozen group cryopreservation was performed according to standard operating procedure at our center. Cryoprotectant solution containing 5% Serum Albumine and 20% dimethylsulfoxyde (DMSO) was slowly added to cellular suspension at 0-4 C° temperature (final DMSO concentration 10%). The cell suspension was than divided into 6-8 cryo-bags, cryopreserved by controlled rate freezing device and finally stored in vapor phase liquid nitrogen cryostorage tank (< -160 °C).”

Point 3: In methods part, in 2.1. Study cohort section it has been stated that "cryopreservation of grafts made according to recommendations from international scientific societies and regulatory authorities", but there are no references.

Response 3: we reported the references (see line 69-71 of the revised manuscript uploaded).

Point 4: One wonders if to analyze the difference between the myeloablative and reduced intensity group, and also between the different ABO groups could provide ant additional information or not.

Response 4: we do not perform this analysis due to the small number of patients.

Point 5: There is a significant p-value for HCT-CL which has not been discussed in discussion part, this should be discussed in relation to other parameters. 

Response 5: thanks, we add a comment on HCT-CL in discussion part (see line 198-200 of the revised manuscript uploaded): Cryo and Fresh groups were well balanced considering baseline characteristics of patients, donors and grafts. Even if proportion of patients with HCT-CL> 2 was higher in Fresh Group (16.7% vs 47.8%, p=0.047), Performace Status was similar in the two cohorts.”

Reviewer 3 Report

This study aimed to evaluate allo-HSCT results after PBSC cryopreservation, which was massively used during COVID pandemic.

To adress this question, authors compared allo-HSCT performed with or without cryopreservation of PBSC from unrelated donors, on a retrospective manner with a monocenter design. 

Results are in line with data previously published, namely in reference number 5. Theses results are well-describer and discussed.

Unfortunately, it seems that no new message could come from these data.  

Nevertheless, results suggest that patients receiving frozen PBSC showed a trend towards less cGVHd, suggesting an effect of cryopreservation on mature T cells alloreactivity. These data sound like a new message which could be more explored. 

In order to reach this goal :

- did the follow-up improve during the submission process, which could permit to reanalyze the cGVHd incidence with a longer follow-up ? 

- performing a propensity score analysis could help to consolidate (or not) the data for cGVHd. 

Author Response

Dear reviewer 3,

thanks for your comments! Here the point-by-piint response

Point 1: did the follow-up improve during the submission process, which could permit to reanalyze the cGVHd incidence with a longer follow-up? 

 Response 1: unfortunately follow-up didn’t improve during the submission process so we are not able to reanalyze the cGVHD incidence for know. We hope to discuss this very interesting point in future with a longer follow up and more biological and clinical details. Thank you for your considerations.

Point 2: performing a propensity score analysis could help to consolidate (or not) the data for cGVHd. 

Response 2: a propensity score analysis was not performed considering the small number of patients that develop a cGVHD.